# Expression of SDF-1/CXCR4 and related inflammatory factors in sodium fluoride-treated hepatocytes

Rui Yang[1,2], Hongting Shen[2], Mingjun Wang[2], Yaqian Zhao[1,2], Shiling Zhu[1,2], Hong Jiang[2], Yanan Li[2], Guanglan Pu[2], Xun Chen[2], Ping Chen[2], Qing Lu[2], Jing Ma[2], Qiang Zhang[2]*

1 Department of Public Health, Medical College, Qinghai University, Xi'ning, China, 2 Department of Endemic Disease Prevention and Control, Qinghai Institute for Endemic Disease Prevention and Control, Xi'ning, China

* wdrr@163.com

**Data Availability Statement:** All relevant data are within the paper and its Supporting information files.

## Abstract

At present, the mechanism of fluorosis-induced damage to the hepatic system is unclear. Studies have shown that excess fluoride causes some degree of damage to the liver, including inflammation. The SDF-1/CXCR4 signaling axis has been reported to have an impact on the regulation of inflammation in human cells. In this study, we investigated the role of the SDF-1/CXCR4 signaling axis and related inflammatory factors in fluorosis through in vitro experiments on human hepatic astrocytes (LX-2) cultured with sodium fluoride. CCK-8 assays showed that the median lethal dose at 24 h was 2 mmol/l NaF, and these conditions were used for subsequent enzyme-linked immunosorbent assays (ELISAs) and quantitative real-time polymerase chain reaction (qPCR) analysis. The protein expression levels of SDF-1/CXCR4 and the related inflammatory factors nuclear factor-κB (NF-κB), interleukin-6 (IL-6), tumor necrosis factor-α (TNF-α) and interleukin 1β (IL-1β) were detected by ELISAs from the experimental and control groups. The mRNA expression levels of these inflammatory indicators were also determined by qPCR in both groups. Moreover, the expression levels of these factors were significantly higher in the experimental group than in the control group at both the protein and mRNA levels ($P < 0.05$). Excess fluorine may stimulate the SDF-1/CXCR4 signaling axis, activating the inflammatory NF-κB signaling pathway and increasing the expression levels of the related inflammatory factors IL-6, TNF-α and IL-1β. Identification of this mechanism is important for elucidating the pathogenesis of fluorosis-induced liver injury.

## 1. Introduction

Fluorosis affects not only the liver system but also other systems of the body. Different types of fluorosis cause different types of damage to the human body, which has been reported in domestic and international studies. Dental fluorosis was observed in some areas of Mexico as early as 1888, but it was not until 1931 that the link between fluoride in drinking water and

**Funding:** The second integrated scientific investigation and research support of the Tibetan Plateau, approval number 2019QZKK0607 The funders had no role in study design, data collection and analysis, decision to publish, or preparation of the manuscript.

**Competing interests:** The authors have declared that no competing interests exist.

dental fluorosis was uncovered [1]. Tooth development is closely related to the functions of enamel-forming cells, and Bronckers [2] concluded that high levels of fluoride can directly affect osteoblasts. In China, three main types of fluorosis are endemic: drinking-water fluorosis, coal-burning fluorosis and tea-drinking fluorosis [3]. There are approximately 60 tea-producing countries in the world, and the number of tea drinkers exceeds 2 billion [4]. In contrast, drinking water fluorosis is still one of the most severe types of fluorosis in China [5]. In highly fluoridated areas, local people have experienced some degree of bone damage due to drinking water with a high fluoride content [6]. Coal-burning fluorosis is also common in China, mainly in the southwest [7, 8]. Approximately 43% of the districts and counties in Guizhou Province, China, were reported to be impacted by fluorosis, and more than half of the individuals in these areas suffer from fluorosis [9]. With global economic development and changes in industry and lifestyle, the prevalence of endemic fluorosis has significantly decreased, but groundwater fluoride contamination and tea-drinking fluorosis are still prevalent in some areas [10–13]. Thus, fluorosis remains a global public health problem that cannot be ignored, and effective control of fluorosis is important for improving quality of life.

Fluorine is an essential trace element that often exists in natural compounds and can be bioactive in humans. The intake of an appropriate amount of fluorine is beneficial, for example, for caries prevention and treatment [14]. Excessive intake of elemental fluoride, however, can cause damage to bones and other important organs in the body [15], such as the cardiovascular and hepatic systems and teeth [16–19]. According to epidemiological investigations and in vitro and in vivo experimental studies, fluorosis may cause a certain degree of damage to the liver, inducing hepatic glucose and lipid metabolic dysfunction, and at the cellular level, fluorosis can cause swelling of the endoplasmic reticulum and mitochondria, reduced nuclear volume, nuclear membrane wrinkling and other defects in hepatocytes [20, 21]. Fluorosis was shown to increase the levels of inflammatory factors [22]. However, the pathogenesis of the resulting liver inflammation is unclear.

Currently, signaling pathways are a hot topic in the study of the pathogenesis of fluorosis. The stromal cell-derived factor 1 (SDF-1) and chemokine receptor 4 (CXCR4) signaling axes are important for regulating inflammation in the human body [23–25]. CXCR4 is a member of the G protein-coupled receptor superfamily and, along with its specific ligand SDF-1, also known as C-X-C motif chemokine ligand 12, forms a chemokine network that is involved in physiological processes such as cellular immunity, inflammatory cell metastasis, and cell proliferation in humans. Chemokines are a large group of small-molecule inflammatory cytokines with molecular weights in the range of 8–12 kDa that act by binding to the corresponding G protein-coupled seven transmembrane receptors present on the surface of target cells to perform various physiological functions [26–28]. The expression of SDF-1/CXCR4 signaling axis members has been reported to be significantly greater in the serum of patients with osteofluorosis than in that of controls, and the expression level of the inflammatory factor NF-κB was positively correlated with chemokine levels [29]. Activation of the NF-κB pathway is closely linked to inflammatory mechanisms in disease [30–32]. The typical NF-κB pathway induces the production of proinflammatory cytokines such as TNF-α and IL-1β in the innate immune system to mediate inflammatory responses. In addition, the SDF-1/CXCR4 signaling axis was shown to activate the NF-κB signaling pathway to participate in the inflammatory response [33–35], suggesting that the SDF-1/CXCR4–NF-κB signaling pathway may play a role in the pathogenesis of fluorosis. Both the SDF-1/CXCR4 signaling axis and the NF-κB signaling pathway can recruit and mobilize inflammatory factors, which may act as signature downstream signaling molecules [36–38]. The mechanism of the SDF-1/CXCR4 signaling axis and its related inflammatory factors NF-κB, IL-6, IL-1β and TNF-α in the context of the pathogenesis of fluorosis has been poorly studied, and differences in the expression of the above

proteins have been reported only in serum samples from individuals in areas with fluorosis [29]. Therefore, the present study aimed to further investigate the role of the CXCL12/CXCR4 signaling axis and the related inflammatory factors IL-1β, TNF-α, IL-6 and NF-κB in the mechanism of fluorosis-induced liver injury at the cellular level.

Although inflammatory damage has not been conclusively established in the pathogenesis of fluorosis, several studies have shown a correlation between these factors. Whether inflammatory factors such as SDF-1/CXCR4, IL-6, TNF-α, NF-κB, and IL-1β play a role in the hepatic system due to fluorosis deserves further exploration.

In this study, LX-2 cells were cultured with sodium fluoride for in vitro experiments, and cell survival was evaluated via the CCK-8 method. Hematoxylin and eosin (HE) staining was used to observe the growth and morphology of the cells in each group, and the expression levels of SDF-1 and CXCR4, as well as those of inflammatory factors related to the SDF-1/CXCR4 signaling pathway, such as IL-6, TNF-α, NF-κB, and IL-1β, were analyzed to determine the mechanism of fluoride intoxication-induced liver inflammation. These results provide a basis for understanding the mechanism of hepatic inflammation caused by fluorosis.

## 2. Materials and methods

### 2.1. Cell culture

Human liver astrocytes (LX-2, 1 vial, model T25) were purchased from Wuhan Procell Life Sciences Co. The cells were cultured in a humidified incubator at 37˚C with 5% $CO_2$ for 48 h. When the cell confluence was greater than 80%, the cells were cultured in a humidified incubator at 37˚C with 5% $CO_2$ and rinsed twice with 2 ml of 1x PBS (Thermo Fisher Scientific, China). Then, 1 ml of trypsin-EDTA (Thermo Fisher Scientific, China) was added to digest the adherent cells, and the cells were placed in a 37˚C and 5% $CO_2$ incubator for 3 min. Then, 4 ml of LX-2 medium was added to the T25 cell culture flasks (Wuhan Procell Life Sciences Co., Ltd., China) to terminate tryptic digestion and mixed with a sterile pasteurized pipette to resuspend the cells. Then, the cells were transferred to 15 ml sterile and enzyme-free centrifuge tubes (Thermo Fisher Scientific, Ltd., China), centrifuged at 1200 × g for 3 min, and resuspended in an appropriate volume of medium for transfer to new culture flasks incubated at 37˚C and 5% $CO_2$ to continue the cultivation of the cell lines. The present study used cells from the 3rd–7th passages.

### 2.2. Sodium fluoride for cell treatments

Sodium fluoride (10.5 g, Tianjin Yong da Chemical Reagent Co., Ltd., China) was dissolved in 500 ml of 1x PBS solution and filtered through a membrane to remove any bacteria, and the concentration was adjusted to 500 mmol/l. Complete media with different NaF concentrations were prepared according to each group, as shown in Table 1. After LX-2 cells were cultured at 37˚C in a 5% $CO_2$ incubator for 24 h, the cells were treated with fluorine (n = 4).

**Table 1. Table of configurations for different NaF concentrations.**

|  | LX-2 special medium content (ml) | Amount of 500 mmol/l NaF added (µl) | Total volume (ml) |
|---|---|---|---|
| 0.5 | 9.99 | 10 | 10 |
| 1 | 9.98 | 20 | 10 |
| 2 | 9.96 | 40 | 10 |
| 4 | 9.92 | 80 | 10 |
| 8 | 9.84 | 160 | 10 |

## 2.3. CCK-8 assay for cell viability

Cells (Nanjing Novozymes Bioscience and Technology Co., Ltd., China) were aliquoted into 96-well plates containing 2,000 cells/ml and incubated for 24 h, after which the culture medium was removed when the cell wall density was greater than 80%. The cells were rinsed twice with PBS, and a blank group, a control group (0 mmol/l NaF), and experimental groups with different concentrations of NaF (0.5, 1, 2, 4, and 8 mmol/l NaF) were prepared. For each well, except for those for the blank group, 200 μl of cell culture medium was added, and the cells were incubated for 12 h, 24 h or 48 h. Then, 10 μl of CCK-8 reagent was added, and the cells were incubated for 1 h in the dark. A fully automated enzyme labeling instrument (Thermo Fisher Scientific, China) was used to determine the OD value of each group at 450 nm, and cell viability (%) was calculated as (OD of the experimental wells-OD of the blank wells/OD of the control wells) × 100% (n = 4).

## 2.4. HE staining

LX-2 cell cultures were fixed with 4% paraformaldehyde for 5–10 min, washed with water for 1 min, treated with HD Constant Staining Pretreatment Solution for 1 min, and sequentially stained with appropriate amounts of hematoxylin and eosin staining solution. Finally, the cells were fixed with neutral resin, and the morphology of LX-2 hepatocytes was observed under a light microscope.

## 2.5. Enzyme-Linked Immunosorbent Assays (ELISAs) of the expression of each target protein in the cell supernatant and cell extract of each group of cells

CXCR4, SDF-1, IL-1β, TNF-α, IL-6 and NF-κB ELISA kits were obtained from Nanjing Boyan Biotechnology Co., Ltd. (China). Cell supernatant collection: Cell supernatants were collected by centrifugation of each test group at 20 min (3000 r/min) and stored at -20°C to avoid repeated freezing and thawing. Cell extract collection: adherent cells were washed with PBS at 4°C and digested with trypsin, and then, the cells were collected after 5 min (1000 r/min) and washed with PBS 3 times. Then, 200 μl of 1x PBS was added to resuspend the cells, the cells were ruptured by repeated freezing and thawing, and the supernatant was collected for evaluation. The expression level of each target protein in the cell supernatant was detected by an enzyme labeling instrument (450 nm). The kits were left at room temperature for 30 min before use. For the wash buffer, the concentrate in the kit was diluted with distilled water at 1:20.

All reagents and samples were used at room temperature. Blank wells (no sample or standard added), standard wells (50 μl of standards at different concentrations) and sample wells (50 μl of samples to be tested) were prepared, and 2 replicate wells were prepared for each group. Horseradish peroxidase (HRP)-labeled antibody (100 μl) was added to each of the wells. The plate was sealed using plate sealing film and incubated for 60 min at 37°C in a water bath or thermostat. The liquid was discarded, 300 μl of wash solution was added to each well, and the plate was washed for 20 s and patted dry on absorbent paper; the washes were repeated 5 times. Then, 50 μl each of chromatography substrates A and B was added to each well and incubated at 37°C for 15 min. Finally, 50 μl of termination solution was added to each well, the color changed from blue to yellow, and readings were taken within 15 min.

## 2.6. RNA extraction

Total RNA was extracted with a Total RNA Extraction Kit (All Specialty Gold Biotechnology Co., Ltd., Beijing, China). Then, cDNA was synthesized with a Reverse Transcription Kit (All

Specialty Gold Biotechnology Co., Ltd., Beijing, China) for gene fragment detection; gene fragments were amplified, and the optimal primers as determined by the HIFI RCR enzyme (All Specialty Gold Biotechnology Co., Ltd., Beijing, China) and HIFI RCR enzyme (Beijing Total Gold Biotechnology Co., Ltd., Beijing, China) were used to amplify the gene fragments at the optimal primer annealing temperature. A fully automated enzyme labeling instrument (Thermo Fisher Scientific) was used to determine the CXCR4, SDF-1, IL-1β, TNF-α, IL-6, and NF-κB mRNA levels as well as the A260/280 ratio. Agarose electrophoresis buffer (LABJIC Biotechnology Co., Ltd., Beijing, China) was used to prepare a 1% agarose gel to determine the RNA concentration.

## 2.7. Real-time quantitative Polymerase Chain Reaction (qPCR)

Each reaction contained 0.4 μl of upstream primer, 0.4 μl of related primer, 0.8 μl of template, 10 μl of 2× PerfectStart Green qPCR SuperMix (Beijing Quan Shi Jin Biotechnology Co., Ltd., Beijing, China), 0.4 μl of Passive Reference Dye (50×), and 8 μl of nuclease-free water for a total volume of 20 μl. Seven pairs of primers were designed by Sangyo Bioengineering Co., Ltd. (Shanghai, China); the primer names, primer sequences, gene IDs, and primer lengths are shown in Table 2. The experimental conditions were 94˚C for 30 s, 59.6˚C for 15 s, and 72˚C for 10 s (44 cycles). The experimental results were obtained by repeating the experiment three times and taking the average value. The mRNA expression of the target genes CXCR4, SDF-1, IL-1β, TNF-α, IL-6 and NF-κB was calculated by the $2^{-\Delta\Delta CT}$ method using glyceraldehyde dehydrogenase 3-phosphate (GAPDH) as the internal reference gene (n = 6).

## 2.8. Statistical analysis

Excel was used for data entry and the development of statistical tables, GraphPad Prism 8.0.2 was used to generate statistical graphs, and SPSS 22.0 was used to statistically analyze the experimental and control groups; a t test was used for data that were normally distributed and exhibited homogeneity of variance, and the Mann−Whitney test was used for data that did not satisfy these conditions. Measurement data are expressed as the mean ± standard deviation (± s), test level α = 0.05, and $P < 0.05$ indicated significance.

**Table 2. Primer sequences, gene numbers and product amplification lengths.**

| Gene | Primer sequences | Product length (bp) | Gene ID |
|---|---|---|---|
| GAPDH | F: GAGGAGGCATTGCTGATGAT | 20 | NM_002046.7 |
| | R: GAAGGCTGGGGCTCATTT | 18 | |
| CXCR4 | F: CTCCTCTTTGTCATCACGCTTCC | 23 | NM_001008540 |
| | R: GGATGAGGACACTGCTGTAGAG | 22 | |
| SDF-1 | F: GGGAAGACCCGTGTTACCAG | 20 | NM_199168.4 |
| | R: AGTCCAGCCTGCTATCCTCA | 20 | |
| NF-κB | F: TGTGTTTGTCCAGCTTCG | 18 | NM_001165412.2 |
| | R: GCTTCTGACGTTTCCTCTG | 19 | |
| IL-1β | F: TCGCCAGTGAAATGATGGCT | 20 | NM_000576 |
| | R: TGAAGCCCTTGCTGTAGTGG | 20 | |
| IL-6 | F: GAGGAGACTTGCCTGGTGAA | 20 | NM_000600.5 |
| | R: CAGCTCTGGCTTGTTCCTCA | 20 | |
| TNF-α | F: ATGAGCACTGAAAGCATGATCC | 22 | NM_000594.4 |
| | R: AGGAGAAGAGGCTGAGGAACAAG | 23 | |

Gene numbers and product amplification lengths

## 3. Results

### 3.1. Effects of different exposure times and sodium fluoride concentrations on the survival of human LX-2 cells

We first investigated the effects of different concentrations of sodium fluoride and different fluoride staining times on cell survival by using CCK-8 assays. The results showed that with different concentrations (0, 0.5, 1, 2, 4 and 8 mmol/l) of NaF, the survival rate of the cells gradually decreased with increasing fluoride concentration, and the LD50 values for LX-2 cells at 12 h, 24 h and 48 h were all approximately 2 mmol/l NaF (see Fig 1). Therefore, 2 mmol/l NaF was selected as the treatment concentration for the subsequent experiments.

The above figure shows the morphology of normal hepatocytes and sodium fluoride-treated LX-2 hepatocytes under a light microscope. Sodium fluoride was added at a concentration of 2 mmol/l for a period of 24 h. The cells in the fluoride-treated group were relatively rounded or swollen; moreover, the nuclei were more centralized, and the cytoplasm was dramatically smaller in volume. Finally, the cellular arrangement was disorganized, and the interstitial space between the cells was enlarged, among other phenomena (see Fig 2).

### 3.2. Increased expression of inflammatory factors in the supernatants of NaF-treated hepatocytes

The OD values of the experimental group and the control group were determined at 450 nm by an enzyme marker, and $t$ tests were used to identify any difference between the OD values of the two groups. ELISA results showed that the expression of CXCR4, SDF-1, IL-1β, TNF-α, IL-6 and NF-κB-specific proteins was significantly increased in the fluoride-treated cell group compared to the control group ($P < 0.05$), as shown in Table 3 and Fig 3.

### 3.3. Increased expression of inflammatory factors in NaF-treated hepatocyte extracts

ELISAs of cell extracts showed that CXCR4, SDF-1 (CXCL12), IL-1β, TNF-α, IL-6, and NF-κB protein expression was significantly increased in the fluoride-treated cell group compared to the control group ($P < 0.05$), as shown in Table 4 and Fig 4.

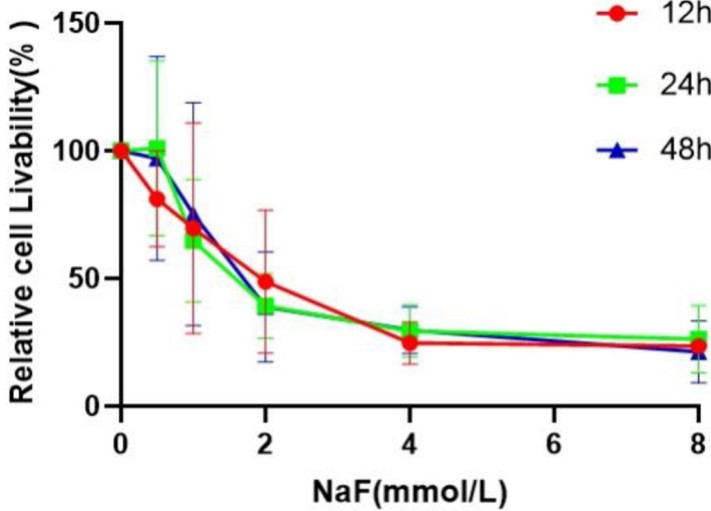

**Fig 1. Effects of sodium fluoride exposure time and concentration on the viability of LX-2 cells determined by CCK-8 assays (*mean± SE*, *n* = 4).**

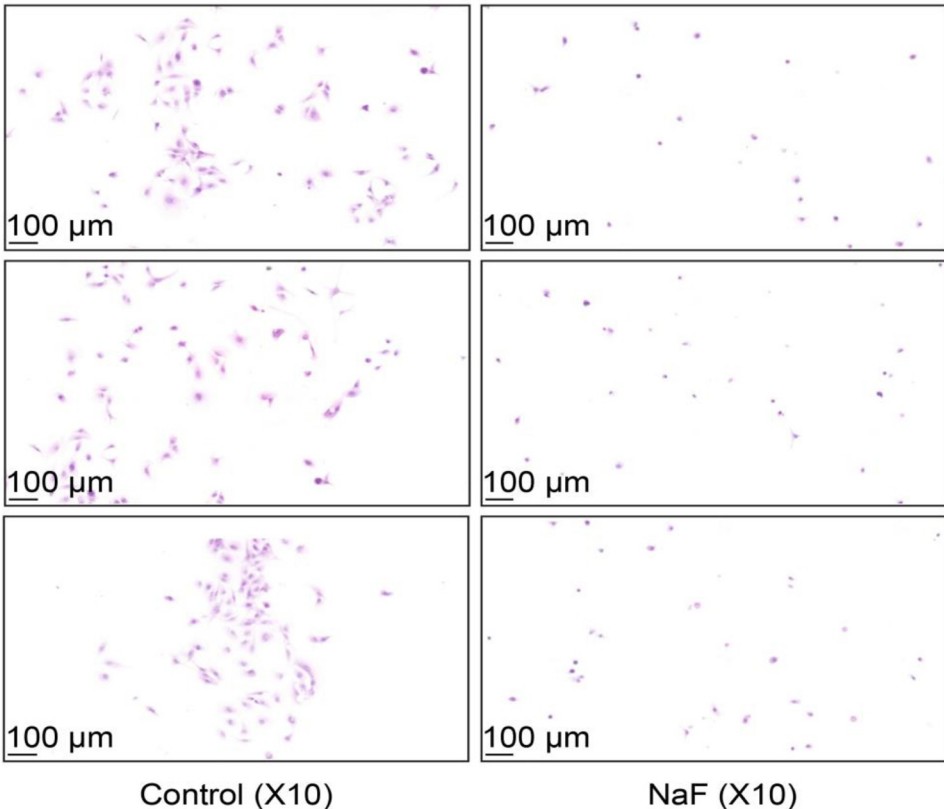

**Fig 2. Results of HE staining of cells (*n* = 3).**

Cell extracts were obtained by repeatedly freezing and thawing the cells, and the lysates were centrifuged to determine whether inflammatory factor receptors were expressed in the cells. The cell supernatants were directly centrifuged and used to determine whether inflammatory factor receptors were expressed on the surface of the cell membranes. ELISAs of both the cell extracts and the cell supernatants showed that fluoride significantly increased the expression of cellular inflammatory factors at this concentration.

### 3.4. Analysis of the qPCR results

Fluorescent dyes were added to the PCR system, and fluorescent signal accumulation was used to monitor the PCR process. We used GAPDH as the internal reference gene, and the 2-ΔΔCT

**Table 3. Expression levels of CXCR4, IL-1β, IL-6, SDF-1, TNF-α and NF-κB in the cell supernatants of the fluoride-treated and control groups ($\bar{x}\pm s$).**

| Testing indicators | CXCR4 (ng/ml) | IL-1β (pg/ml) | IL-6 (pg/ml) | SDF-1 (ng/ml) | TNF-α (pg/ml) | NF-κB (pg/ml) |
|---|---|---|---|---|---|---|
| Number of samples in the fluoride group (*n*) | 4 | 4 | 4 | 4 | 4 | 4 |
| Sample size of the control group (*n*) | 4 | 4 | 4 | 4 | 4 | 4 |
| Indicator expression in the fluoride-treated groups | 2.13±0.07[a] | 15.5±0.29 [a] | 28.61±2.55[a] | 2.2±0.12[a] | 13.52±0.27[a] | 415.40±14.72[a] |
| Expression of indicator in the control group | 1.79±0.04 | 13.37±0.09 | 9.84±0.49 | 1.68±0.04 | 11.55±0.51 | 295.28±11.31 |
| *t* | 4.30 | 6.97 | 7.24 | 4.17 | 3.40 | 6.47 |
| *P* | <0.05 | <0.05 | <0.05 | <0.05 | <0.05 | <0.05 |

Note:

[a] *P* < 0.05 compared to the same indicator in the control group the *t* value is from a two-sample *t* test, n = 4

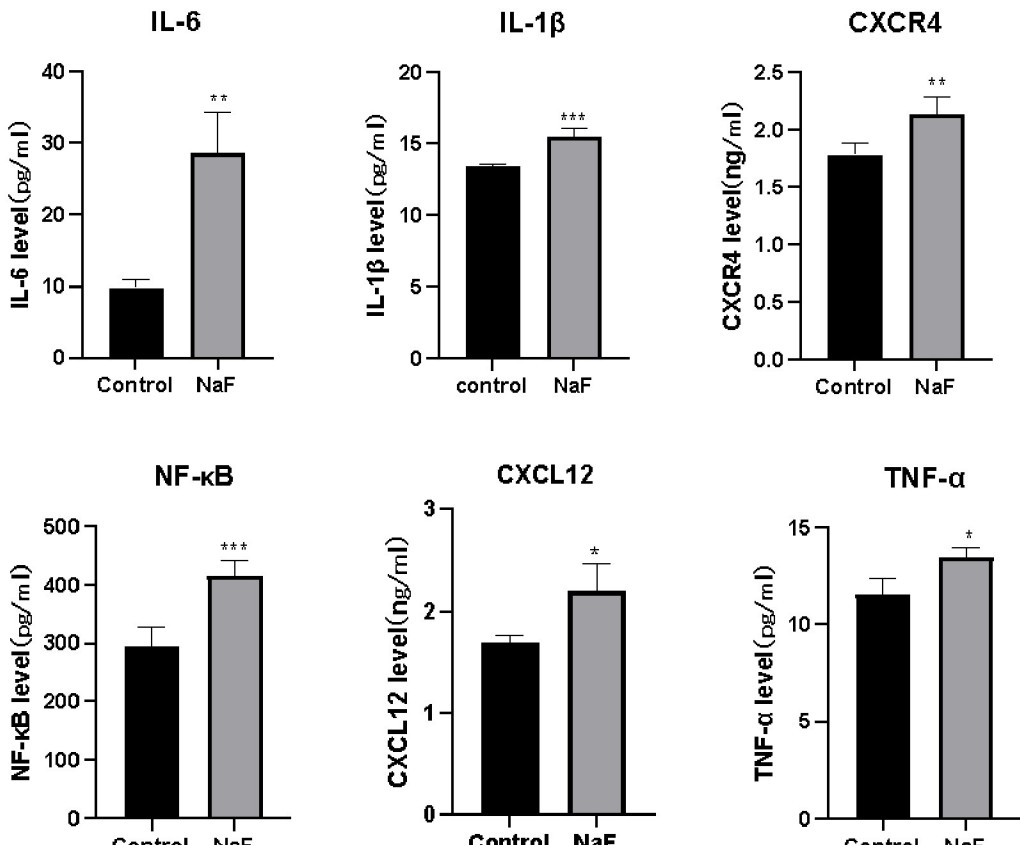

**Fig 3. Expression level of each target protein in the supernatants of the fluoride-treated and control cells (*mean±SE*, *n* = 4).** *P < 0.05, **P < 0.01, and ***P < 0.001 compared with the control group.

method was used to calculate the mRNA levels of CXCR4, SDF-1, IL-1β, TNF-α, IL-6 and NF-κB, in the fluoride group and the control group, as shown in Fig 5.

We concluded that fluoride enhanced the SDF-1/CXCR4 signaling axis and the inflammatory factors IL-1β, TNF-α, IL-6 and NF-κB in normal hepatocytes, both in terms of gene expression and protein expression, as shown by ELISA and qPCR.

## 4. Discussion

Fluorosis occurs in many regions of the world and affects human quality of life to a certain extent not only by placing an economic burden on families but also by impeding the economic

**Table 4. Expression levels of SDF-1, TNF-α, NF-κB, CXCR4, IL-1β and IL-6 in cell extracts from the fluoride-treated and control groups ($\bar{x}±s$).**

| Testing indicators | SDF-1 (ng/ml) | TNF-α (pg/ml) | NF-κB (pg/ml) | CXCR4 (ng/ml) | IL-1β (pg/ml) | IL-6 (pg/ml) |
|---|---|---|---|---|---|---|
| Number of samples in the fluoride group (*n*) | 4 | 4 | 4 | 4 | 4 | 4 |
| Sample size of the control group (*n*) | 4 | 4 | 4 | 4 | 4 | 4 |
| Indicator expression in the fluoride-contaminated groups | 5.08±0.32[a] | 41.99±1.91[a] | 713.18±39.38[a] | 5.74±0.38[a] | 65.19±1.76 [a] | 35.49±1.82[a] |
| Expression of indicator in the control group | 3.94±0.21 | 52.10±3.33 | 523.17±36.27 | 4.28±0.37 | 46.93±1.80 | 25.47±2.50 |
| *t* | 3.00 | 2.64 | 3.55 | 2.73 | 7.23 | 3.25 |
| *P* | <0.05 | <0.05 | <0.05 | <0.05 | <0.05 | <0.05 |

Note:

[a] *P* < 0.05 compared to the same indicator in the control group the *t* value is a two-sample *t* test, n = 4

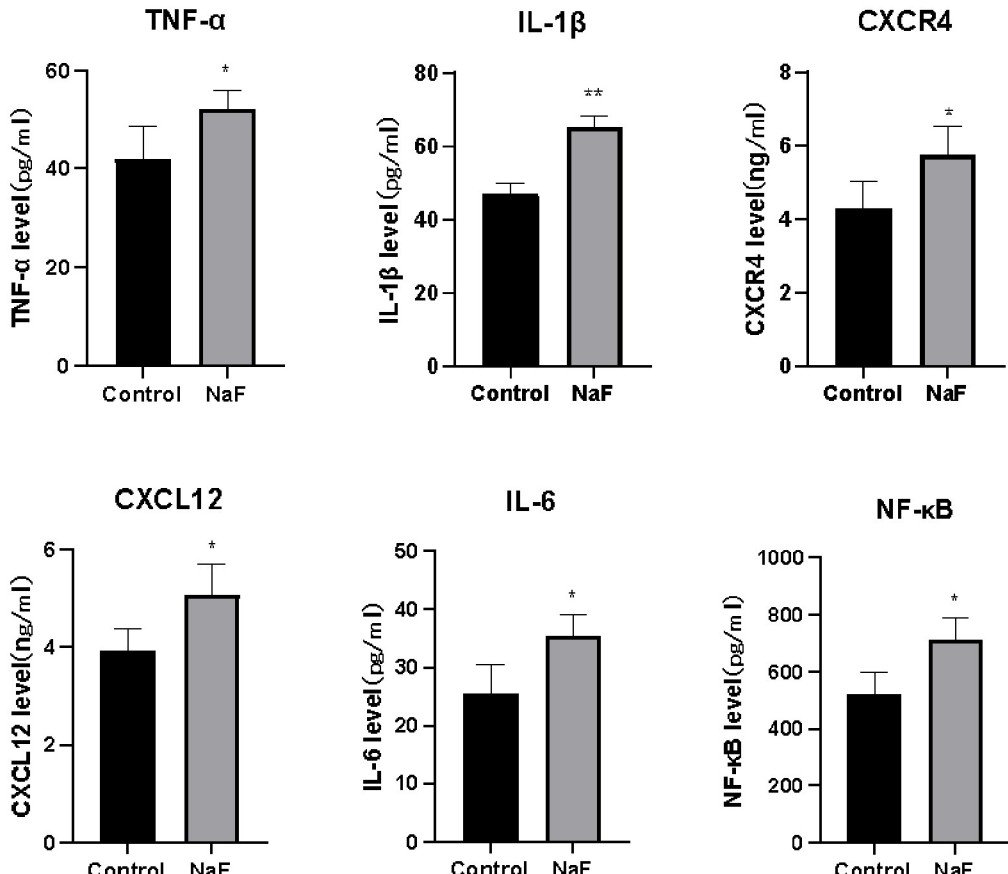

**Fig 4. Expression level of each target protein in cell extracts from the fluoride-treated and control cells** (*mean±SE*, *n* = 4). *$P < 0.05$ and **$P < 0.01$ compared with the control group.

development of a region [39]. However, the pathogenesis of fluorosis has not yet been determined, although fluorosis is known to involve mainly oxidative stress, endoplasmic reticulum stress, and signaling pathways that can damage human tissues or organs [40–42]. Although inflammatory damage is not widely recognized as a major component of fluorosis, fluorosis has been shown to cause increased expression of inflammatory factors such as IL-6 [43, 44]. Thus, inflammatory damage may be one of the pathogenic mechanisms leading to fluorosis, but whether this damage plays a role in specific cases needs to be evaluated on a case-by-case basis. SDF-1/CXCR4, as important signaling pathways, recruits cellular immune factors to participate in the body's inflammatory response [45, 46]. In the present study, human LX-2 cells were treated with different concentrations of NaF in vitro, and the cell survival rate was detected via the CCK-8 method. The LD50 value, which was found to be 2 mmol/l after fluoridation with different concentrations of NaF (0.5, 1, 2, 4, and 8 mmol/l) for 12, 24, and 48 h, was selected as the appropriate concentration for further experiments. We analyzed the morphology of the cells before and after fluoride treatment, as well as the mRNA and protein expression of each target, CXCR4, CXCL12, NF-κB, IL-6, IL-1β and TNF-α. One limitation of the present study is that we did not explore the relationship between the signaling axis and inflammatory factors, which needs to be explored by signaling pathway antagonist experiments. However, the present cellular experiments successfully identified the SDF-1/CXCR4 signaling axis, and the related inflammatory factors NF-κB, IL-6, IL-1β, and TNF-α were

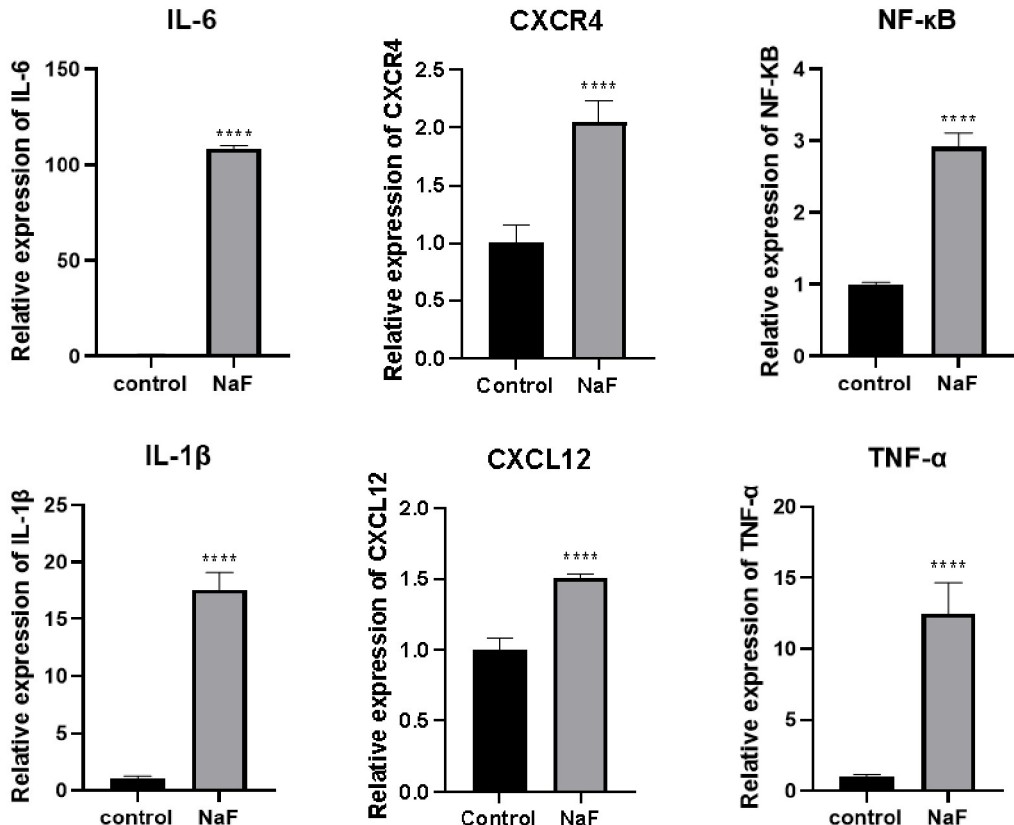

**Fig 5. mRNA expression levels of the indicators in the fluoride group and the control group (****$P < 0.001$, *mean* $\pm SE$, $n = 6$).**

expressed at high levels in hepatocytes treated with high concentrations of fluoride. We found that the gene and protein expression levels of SDF-1/CXCR4 signaling axis members in the fluoride group were significantly greater than those in the blank control group, and the mRNA expression levels of inflammatory molecules such as NF-κB, IL-6, IL-1β, and TNF-α were consistent with those of the SDF-1/CXCR4 signaling axis in both groups. We speculated that fluoride could lead to upregulated expression of genes and proteins in the SDF-1/CXCR4 signaling axis in hepatocytes and subsequently activate the NF-κB signaling pathway to release excessive inflammatory factors such as IL-6, IL-1β and TNF-α. The association between the pathogenesis of inflammation in fluorosis and signaling pathways may be confirmed by further investigating signaling pathway antagonists that block signaling pathways to explore the expression of inflammatory factors. As the factors that cause inflammation are very complex, the present findings may also suggest synergistic activation of the NF-κB signaling pathway and other signaling pathways, which are involved in the release of the inflammatory factors IL-6, IL-1β, and TNF-α [47]. However, the underlying mechanism needs to be further studied.

## 5. Conclusion

Excessive sodium fluoride induced an increase in the expression of the hepatic cellular inflammatory factors IL-6, TNF-α and IL-1β as well as the chemokine signaling axis SDF-1/CXCR4 and the inflammatory signaling pathway NF-κB.

## Supporting information

**S1 Data.**
(XLSX)

## Author Contributions

**Conceptualization:** Rui Yang.

**Data curation:** Hong Jiang, Yanan Li.

**Investigation:** Guanglan Pu.

**Resources:** Qiang Zhang.

**Supervision:** Yaqian Zhao, Shiling Zhu.

**Validation:** Hongting Shen.

**Visualization:** Mingjun Wang, Xun Chen, Ping Chen, Qing Lu, Jing Ma.

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
