## [Decision Letter · Decision Letter 0]

4 Dec 2023

PONE-D-23-32257Expression of SDF-1/CXCR4 and Related Inflammatory Factors in Sodium Fluoride-Treated HepatocytesPLOS ONE

Dear Dr. Zhang,

Thank you for submitting your manuscript to PLOS ONE. After careful consideration, we feel that it has merit but does not fully meet PLOS ONE’s publication criteria as it currently stands. Therefore, we invite you to submit a revised version of the manuscript that addresses the points raised during the review process.

We look forward to receiving your revised manuscript.

Kind regards,

Vara Prasad Saka

Academic Editor

PLOS ONE

“Conceptualization, R.Y.; methodology, R.Y.; software, R.Y.; validation, R.Y. and H.S.; formal analysis,R.Y. and H.S.; investigation, R.Y., G.P. and R.Y.; data curation, Y.L., J.H. and M.W.; writing—original draft preparation, R.Y; writing—review and editing, R.Y.; visualization, R.Y., X.C. and P.C. and J.M. and L.Q.; supervision,R.Y. and Y.Z. and S.Z.; project administration, Q.Z.; funding acquisition, Q.Z. All authors have read and agreed to the published version of the manuscript.”

“The authors thank the Institute of Geographic Sciences and Resources of the Chinese Academy of Sciences for financial support.”

“Conceptualization, R.Y.; methodology, R.Y.; software, R.Y.; validation, R.Y. and H.S.; formal analysis,R.Y. and H.S.; investigation, R.Y., G.P. and R.Y.; data curation, Y.L., J.H. and M.W.; writing—original draft preparation, R.Y; writing—review and editing, R.Y.; visualization, R.Y., X.C. and P.C. and J.M. and L.Q.; supervision,R.Y. and Y.Z. and S.Z.; project administration, Q.Z.; funding acquisition, Q.Z. All authors have read and agreed to the published version of the manuscript.”

7. PLOS requires an ORCID iD for the corresponding author in Editorial Manager on papers submitted after December 6th, 2016. Please ensure that you have an ORCID iD and that it is validated in Editorial Manager. To do this, go to ‘Update my Information’ (in the upper left-hand corner of the main menu), and click on the Fetch/Validate link next to the ORCID field. This will take you to the ORCID site and allow you to create a new iD or authenticate a pre-existing iD in Editorial Manager. Please see the following video for instructions on linking an ORCID iD to your Editorial Manager account: https://www.youtube.com/watch?v=_xcclfuvtxQ.

8. Please amend the manuscript submission data (via Edit Submission) to include author Rui Yang.

9. Please amend your authorship list in your manuscript file to include author 瑞 杨.

Reviewers' comments:

Reviewer's Responses to Questions

**Comments to the Author**

1. Is the manuscript technically sound, and do the data support the conclusions?

Reviewer #1: Yes

Reviewer #2: Yes

Reviewer #3: Yes

2. Has the statistical analysis been performed appropriately and rigorously? 

Reviewer #1: Yes

Reviewer #2: Yes

Reviewer #3: Yes

3. Have the authors made all data underlying the findings in their manuscript fully available?

Reviewer #1: Yes

Reviewer #2: Yes

Reviewer #3: Yes

4. Is the manuscript presented in an intelligible fashion and written in standard English?

Reviewer #1: Yes

Reviewer #2: Yes

Reviewer #3: Yes

5. Review Comments to the Author

Reviewer #1: Dear correspondence author

Please find your manuscript with few comments. Kindly be sure to correct them all to improve your article.

The elimination of fluoride from the human body primarily occurs through urine, feces, and sweat. The elimination half-life of fluoride in humans (in vivo) varies depending on factors such as age, kidney function, and overall health. In general, the elimination half-life of fluoride ranges from a few hours to several days. That makes your incubation times (12, 24 and 48 h) and concentrations (0.5, 1, 2, 4, 8 mmol NaF /L) looks logic.

Meanwhile, I have some interrogations:

1- What is the source of the human hepatic astrocytes (LX-2)? Did you obtain it from Wuhan Procell Life Sciences Co? There is a contradictory between the sources in page 8th first line!!

2- In histopathology figure, I can't see all the details you mentioned, please check the brightness and or the contrast. Also, I didn’t know the differences between the first, second and third rows?

3- You said in statistical analysis that you expressed the mean plus minus standard deviation not standard errors, while in page 19 you mentioned “mean plus minus standard errors in figure 5”. Please declare which is the right?

4- I strongly recommend you to delete repeating your results in the “discussion” section and go deeper in explaining, analyzing, and interpreting your findings in the light of mitochondrial dysfunction and link it to the inflammatory molecules in your study.

Wish you the best luck

Reviewer #2: The manuscript expresses the work which is technically sound and presented in an intelligible fashion. The data presented is sufficient in deriving the conclusion of the work and can be accepted in the present form.

Reviewer #3: The authors have studied the “Expression of SDF-1/CXCR4 and Related Inflammatory Factors in Sodium Fluoride-Treated Hepatocytes”. This research article is well-designed and interesting. The manuscript is suitable for publication but there are minor technical flaws which need to be addressed. Some of these concerns are mentioned below:

1- Why were just LX-2 astrocytes used in the study? Briefly explain it.

2- The manuscript does not include a reference to the cell culture methodology used.

3- The apparatus used for microscopy throughout the study is not mentioned.

4- The source from which the primers were chosen is unavailable. Give the source in manuscript.

5- There is a lack of referencing in the entire methodology. For the methods section, not a single reference is provided.

6- In photography, the scale is not indicated.

7- It is advised that a higher magnification image of HE staining of cells be included because there is no evident comparison with the images provided with the current file.

8- The outcome is repeated in the discussion section. It should be modified.

9- The significance of the current investigation is not clearly stated in the text. What is the relevance of the study? Describe it.

10- Is there any recent research on fluorosis with inflammation? If yes, explain why the current investigation is necessary.

6. PLOS authors have the option to publish the peer review history of their article (what does this mean?). If published, this will include your full peer review and any attached files.

Reviewer #1: **Yes: **Waleed Fathy Khalil

Reviewer #2: **Yes: **GV Narasimha Kumar

Reviewer #3: No

---

## [Author Response · Author response to Decision Letter 0]

19 Feb 2024

1.Why were just LX-2 astrocytes used in the study? Briefly explain it.

Firstly, LX-2 cells are easy to culture, grow vigorously and require less conditions such as culture medium, and secondly, the inflammatory factors as well as signaling axes involved in this study are less studied in LX-2 cells, which is of great significance for the study of the liver system due to fluorosis.

2.The manuscript does not include a reference to the cell culture methodology used.

I have already mentioned the cell culture process in Materials and Methods under Cell Culture and Transmission

3.The apparatus used for microscopy throughout the study is not mentioned.

The microscope mentioned in the article is a general optical microscope, which is used directly for observing the state of cell growth as well as HE sections, and I don't feel that the article needs to go into further detail about the specific instrumentation involved in that general microscope. Perhaps I misunderstood what the author meant by specific instruments, I don't quite understand it!

4.The source from which the primers were chosen is unavailable. Give the source in manuscript.

I have already mentioned in my manuscript that my primers are from Shanghai Sangong Bioengineering Co. in China

5.There is a lack of referencing in the entire methodology. For the methods section, not a single reference is provided. 

The ELISA, PCR, HE staining, etc. covered in the article I don't feel the need to provide appropriate references for this, they are just one method

6.In photography, the scale is not indicated.

I have labeled the HE stained images with the scale you requested.

7.It is advised that a higher magnification image of HE staining of cells be included because there is no evident comparison with the images provided with the current file.

Dear Sir, I have tried to make the cell HE staining pictures as clear as possible, mainly by comparing the number of cells and cell morphology of the two groups, thank you for your valuable comments!

8.The outcome is repeated in the discussion section. It should be modified.

This section is currently redacted.

9.The significance of the current investigation is not clearly stated in the text. What is the relevance of the study? Describe it.

The introduction and conclusion of the article have actually explained the significance of the study, mainly through the sodium fluoride infiltration of human LX-2 hepatocytes, and then explore the signaling pathway and the expression level of related inflammatory factors, in order to provide ideas for the further exploration of fluorosis-induced pathogenesis of the hepatic system

10.Is there any recent research on fluorosis with inflammation? If yes, explain why the current investigation is necessary.

The recognized doctrines of fluorosis are oxidative stress, signaling pathways and other doctrines. The theory of inflammatory response to fluorosis has not been established, and domestic and foreign research is still in the exploratory stage, with relatively little research content. Therefore, it is slightly important to study the inflammatory response from the perspective of the liver pathogenesis of fluorosis to provide new ideas for the prevention and treatment of fluorosis.

---

## [Editor Report · Decision Letter 1]

9 Apr 2024

Expression of SDF-1/CXCR4 and Related Inflammatory Factors in Sodium Fluoride-Treated Hepatocytes

PONE-D-23-32257R1

Dear Dr. Zhang,

We’re pleased to inform you that your manuscript has been judged scientifically suitable for publication and will be formally accepted for publication once it meets all outstanding technical requirements.

Kind regards,

Vara Prasad Saka

Academic Editor

PLOS ONE
---

## [Editor Report · Acceptance letter]

13 May 2024

PONE-D-23-32257R1 

PLOS ONE

Dear Dr. Zhang, 

I'm pleased to inform you that your manuscript has been deemed suitable for publication in PLOS ONE. Congratulations! Your manuscript is now being handed over to our production team.

Kind regards, 

on behalf of

Dr. Vara Prasad Saka 

Academic Editor

PLOS ONE